# A Novel ML-Aided Methodology for SINS/GPS Integrated Navigation Systems during GPS Outages

**Jin Sun** [1,2,*] **, Zhengyu Chen** [3] **and Fu Wang** [4]

1. School of Internet of Things, Nanjing University of Posts and Telecommunications, Nanjing 210003, China
2. State Key Laboratory of Ocean Engineering, Shanghai Jiao Tong University, Shanghai 200240, China
3. College of Automation and College of Artificial Intelligence, Nanjing University of Posts and Telecommunications, Nanjing 210003, China
4. East China Institute of Photo-Electron IC, Suzhou 215129, China
* Correspondence: sunjin@njupt.edu.cn

**Abstract:** To improve the navigation accuracy for land vehicles during global positioning system (GPS) outages, a machine learning (ML) aided methodology to integrate a strap-down inertial navigation system (SINS) and GPS system is proposed, as follows. When a GPS signal is available, an online sequential extreme learning machine with a dynamic forgetting factor (DOS-ELM) algorithm is used to train the mapping model between the SINS' acceleration, specific force, speed/position increments outputs, and the GPS' speed/position increments. When a GPS signal is unavailable, GPS speed/velocity measurements are replaced with prediction output of the well-trained DOS-ELM module's prediction output, and information fusion with the SINS reduces the degree of system error divergence. A land vehicle field experiment's actual sensor data were collected online, and the DOS-ELM-aided methodology for the SINS/GPS integrated navigation systems was applied. The simulation results indicate that the proposed methodology can reduce the degree of system error divergence and then obtain accurate and reliable navigation information during GPS outages.

**Keywords:** global positioning system (GPS) outages; strap-down inertial navigation system (SINS); machine learning (ML); online sequential extreme learning machine with dynamic forgetting factor (DOS-ELM); Kalman filtering (KF); SINS/GPS integrated systems



## 1. Introduction

To provide global and all-weather navigation information, a strap-down inertial navigation system (SINS) relies only on a gyroscope and an accelerometer to sense the motion of the carrier in the inertial frame; it is an independent and autonomous navigation system. It has outstanding advantages, such as continuous output of the carrier's position, speed, and attitude information, high short-term navigation accuracy, and complete independence [1]. It is widely used in vehicle, ship, aircraft, tactical, and strategic navigation.

With the development of the global navigation satellite system (GNSS), represented by a global positioning system (GPS), a SINS can provide high-precision global and all-weather navigation and positioning services in which error does not accumulate over time. Satellite navigation systems have played essential roles in military, aviation, economic construction, and scientific fields [2]. However, because satellite signals are easily blocked and experience regular interference, the data update rate of the navigation result is low, and there is no attitude information output. Combined with the advantages of the SINS (short-term positioning accuracy and high data sampling rate), the integrated navigation algorithm can obtain the three-dimensional position, speed, stability, and reliability attitudes, good accuracy, and a high data update rate. The integrated navigation algorithm facilitates the complementary functioning of SINS and GPS systems [3]. First, inertial navigation

results are corrected using satellite navigation results in which errors do not accumulate over time; thus, avoiding the rapid accumulation of errors over time. Second, high-precision and high-stability inertial navigation results in a short period can partially solve the navigation and positioning problems when the satellite signal is blocked. The integrated navigation systems improve the robustness of navigation results. In addition, integrated navigation can estimate the constant errors of the inertial element and feedback, and correct the accelerometer and gyroscope outputs to realize the online calibration of the inertial element [4]. Therefore, the integration of satellite navigation and inertial navigation can obtain stable and reliable three-dimensional position, speed, and attitude information with good accuracy and a high data update rate.

Although integrated navigation systems can provide navigation information to users in most locations on Earth, doing so requires capturing standard satellite signals. In sheltered outdoor areas, such as cities, canyons, and forests, when satellite signals are attenuated or lost due to occlusion by buildings, mountains, trees, etc., the errors of using a pure inertial navigation system (INS) accumulates rapidly over time. This results in decreased positioning accuracy and an inability to navigate properly [5]. In rapidly changing cities, there are many large floors with dense forests, and an increasing number of large and sealed indoor environments; as a result, the application of satellite and integrated navigation systems in complex environments is minimal.

For the past few years, the rapid development of machine learning (ML) technology has led many researchers to begin employing ML-aided SINS/GPS integrated navigation systems to improve the SINS' navigation performance during GPS outages [6–21]. The specific working principle is that when the GNSS's signal is available, an ML algorithm trains the mapping model between the SINS' acceleration, specific force, speed, and position increments outputs and GNSS's speed/position increments [6–13]. When GNSS data are unavailable, the GPS' speed/velocity measurements are replaced with the well-trained ML module's prediction outputs; information fusion with the SINS reduces the degree of system error divergence [14–21]. ML technology has an excellent capability to learn and reason in an inaccurate and uncertain environment, and in this way it corresponds to the human brain. It can effectively compensate for the inherent flame of the traditional Kalman filtering (KF) theory when integrated navigation data are fused. Even during GPS outages, ML technology can aid KF in forecasting and estimating the navigation calculation error of the SINS; the accuracy of the integrated system is improved through error compensation. However, these ML methods are unsuitable for processing data streams in online learning scenarios.

Therefore, this paper proposes an online sequential extreme learning machine with a dynamic forgetting factor (DOS-ELM) aided methodology for a SINS/GPS integrated navigation system during GPS outages is proposed. The main contributions of this paper are summarized as follows: (1) The DOS-ELM algorithm is used to train the mapping model between the SINS' acceleration, specific force, speed, and position increments outputs and the GPS' speed/position increments. When the GPS' signal is unavailable, GPS speed/velocity measurements are replaced with the well-trained DOS-ELM module's prediction outputs, and information fusion with the SINS reduces the degree of system error divergence. A semi-physical simulation was performed to verify the feasibility and effectiveness of the proposed methodology. (2) Each time the proposed model is updated, the DOS-ELM algorithm can adjust the forgetting factor according to the difference between the prediction accuracies of the current and previous models. Thus, the model can dynamically adjust the relative importance of contemporary and historical data according to changes in the data flow. This allows the model to adapt faster and more accurately to the current environment.

The outline of this paper is as follows. After this introduction, Section 2 establishes the KF model of the SINS/GPS loosely integrated navigation systems. Section 3 presents the design process for the proposed DOS-ELM-aided methodology for SINS/GPS integrated navigation systems during GPS outages. The actual vehicle-mounted experimental data is used for semi-physical simulation in Section 4. Finally, Section 5 concludes.

## 2. KF Model of SINS/GPS Integrated Navigation Systems

Position and speed measurements are used in SINS/GPS loosely integrated navigation systems. The difference between the position/speed obtained by the GPS receiver and the position/speed calculated by the SINS are directly used as Kalman filter inputs. The Kalman filter output adopts feedback correction; the drift error correction of the gyroscope and accelerometer are corrected in SINS. In contrast, the position and speed information directly correct the SINS's calculation results. Thus, the integrated method's advantages are its simple structure and easily implemented engineering. The two navigation subsystems are independent, and the navigation information has a certain degree of redundancy.

The state equation of the SINS/GPS integrated navigation systems is as follows:

$$\dot{X}(t) = F(t)X(t) + G(t)W(t) \tag{1}$$

where $X(t)$ is the state vector; it is as follows:

$$X(t) = [\delta V_E \, \delta V_N \, \delta V_U \, \phi_E \, \phi_N \, \phi_U \, \delta L \, \delta\lambda \, \delta h \, \nabla_x \nabla_y \nabla_z \, \varepsilon_x \varepsilon_y \varepsilon_z]^T \tag{2}$$

where $\delta V$ denotes the SINS speed error, and $\delta V = [\delta V_E \quad \delta V_N \quad \delta V_U]^T$. $E, N$, and $U$ are the eastward, northward, and upward axes of the n coordinate system, respectively. $\phi$ represents the attitude error, and $\phi = [\phi_E \quad \phi_N \quad \phi_U]^T$. $\delta L, \delta\lambda$ and $\delta h$ are the system's latitude, longitude, and altitude errors, respectively. $\nabla_b$ denotes the accelerometer bias, and $\nabla_b = [\nabla_x \quad \nabla_y \quad \nabla_z]^T$. $x, y$, and $z$ are the three axes of the b coordinate system, respectively. $\varepsilon^b$ is the gyroscope drift, and $\varepsilon^b = [\varepsilon_x \quad \varepsilon_y \quad \varepsilon_z]^T$.

The system noise vector $W(t)$ is as follows:

$$W(t) = [\omega_x \, \omega_y \omega_z \, a_x \, a_y \, a_z]^T \tag{3}$$

where $\omega_x$, $\omega_y$, and $\omega_z$ represent the noise of the x axial, y axial, and z axial gyroscope, respectively. $a_x$, $a_y$, and $a_z$ represent the noise of the x axial, y axial, and z axial accelerometers, respectively. Their means are all zero, and they obey the standard Gaussian distribution.

The system state transition matrix $F(t)$ can be derived according to the attitude error equation, velocity error equation, and position error equation [1,2]; it is as follows:

$$
F(t) = \begin{bmatrix}
F_{11} & F_{12} & F_{13} & 0 & -f_U & f_N & F_{17} & 0 & F_{19} & C_{11} & C_{12} & C_{13} & 0 & 0 & 0 \\
F_{21} & -\dfrac{V_U}{R_N} & -\dfrac{V_E}{R_N} & f_U & 0 & -f_E & F_{27} & 0 & F_{29} & C_{21} & C_{22} & C_{23} & 0 & 0 & 0 \\
F_{31} & \dfrac{2V_N}{R_N} & 0 & -f_N & f_E & 0 & F_{37} & 0 & F_{39} & C_{31} & C_{32} & C_{33} & 0 & 0 & 0 \\
0 & -\dfrac{1}{R_N} & 0 & 0 & F_{45} & F_{46} & 0 & 0 & 0 & 0 & 0 & 0 & -C_{11} & -C_{12} & -C_{13} \\
\dfrac{1}{R_E} & 0 & 0 & F_{54} & 0 & F_{56} & F_{57} & 0 & -\dfrac{V_E}{R_E^2} & 0 & 0 & 0 & -C_{21} & -C_{22} & -C_{23} \\
\dfrac{\tan L}{R_E} & 0 & 0 & F_{64} & F_{65} & 0 & F_{67} & 0 & F_{69} & 0 & 0 & 0 & -C_{31} & -C_{32} & -C_{33} \\
0 & \dfrac{1}{R_N} & 0 & 0 & 0 & 0 & 0 & 0 & -\dfrac{V_N}{R_N^2} & 0 & 0 & 0 & 0 & 0 & 0 \\
\dfrac{1}{\cos L R_N} & 0 & 0 & 0 & 0 & 0 & F_{87} & 0 & F_{89} & 0 & 0 & 0 & 0 & 0 & 0 \\
0 & 0 & 1 & 0 & 0 & 0 & 0 & 0 & 0 & 0 & 0 & 0 & 0 & 0 & 0 \\
0 & 0 & 0 & 0 & 0 & 0 & 0 & 0 & 0 & 0 & 0 & 0 & 0 & 0 & 0 \\
0 & 0 & 0 & 0 & 0 & 0 & 0 & 0 & 0 & 0 & 0 & 0 & 0 & 0 & 0 \\
0 & 0 & 0 & 0 & 0 & 0 & 0 & 0 & 0 & 0 & 0 & 0 & 0 & 0 & 0 \\
0 & 0 & 0 & 0 & 0 & 0 & 0 & 0 & 0 & 0 & 0 & 0 & 0 & 0 & 0 \\
0 & 0 & 0 & 0 & 0 & 0 & 0 & 0 & 0 & 0 & 0 & 0 & 0 & 0 & 0 \\
0 & 0 & 0 & 0 & 0 & 0 & 0 & 0 & 0 & 0 & 0 & 0 & 0 & 0 & 0
\end{bmatrix}
\tag{4}
$$

where $C_{ij}(i, j = 1, 2, 3)$ is the element of the attitude matrix $C_b^n$. $V$ denotes the SINS speed, and $V^n = \begin{bmatrix} V_E & V_N & V_U \end{bmatrix}^T$. $f$ indicates the acceleration, and $f^n = \begin{bmatrix} f_E & f_N & f_U \end{bmatrix}^T$. $\omega_{ie}$ is the Earth's rotation angular speed. $R_E$ and $R_N$ are the semi-major axis and radius of curvature along the circle of the Earth, respectively. $L$ is the geographic latitude. Other elements in the matrix $F(t)$ are represented as follows:

$$
F_{11} = \frac{V_N}{R_E}\tan L - \frac{V_U}{R_E},\ F_{12} = 2\omega_{ie}\sin L + \frac{V_E}{R_E}\tan L,\ F_{13} = -2\omega_{ie}\cos L - \frac{V_E}{R_E},
$$
$$
F_{17} = 2\omega_{ie}(V_U \sin L + V_N \cos L) + \frac{V_E V_N}{R_E}\sec^2 L,\ F_{19} = \frac{V_E V_U - V_E V_N \tan L}{R_E^2},
$$
$$
F_{21} = -2\left(\omega_{ie}\sin L + \frac{V_E}{R_E}\tan L\right),\ F_{27} = -\left(2\omega_{ie}\cos L + \frac{V_E}{R_E}\sec^2 L\right)V_E,\ F_{29} = \frac{V_N V_U}{R_N^2} + \frac{V_E^2 \tan L}{R_E^2},
$$
$$
F_{31} = 2\left(\omega_{ie}\cos L + \frac{V_E}{R_E}\right),\ F_{37} = -2\omega_{ie}V_E \sin L,\ F_{39} = -\left(\frac{V_E^2}{R_E^2} + \frac{V_N^2}{R_N^2}\right),\ F_{45} = \omega_{ie}\sin L + \frac{V_E}{R_E}\tan L,
$$
$$
F_{46} = -\left(\omega_{ie}\cos L + \frac{V_E}{R_E}\right),\ F_{54} = -\left(\omega_{ie}\sin L + \frac{V_E}{R_E}\tan L\right),\ F_{56} = -\frac{V_N}{R_N},\ F_{57} = -\omega_{ie}\sin L,
$$
$$
F_{64} = \left(\omega_{ie}\cos L + \frac{V_E}{R_E}\right),\ F_{65} = \frac{V_N}{R_N},\ F_{67} = \omega_{ie}\cos L + \frac{V_E}{R_E}\sec^2 L,\ F_{69} = -\frac{V_E \tan L}{R_E^2},
$$
$$
F_{87} = \frac{V_E}{R_E}\sec^2 L \sin L,\ F_{89} = -\frac{V_E \sec L}{R_E^2}.
$$

The system process noise transfer matrix is as follows:

$$
G(t) = \begin{bmatrix}
C_b^n & 0_{3\times 3} \\
0_{3\times 3} & C_b^n \\
0_{9\times 3} & 0_{9\times 3}
\end{bmatrix}
\tag{5}
$$

The system measurement equation is as follows:

$$
Z(t) = H(t)X(t) + V(t)
\tag{6}
$$

where $H(t)$ is the observation vector, and $V(t)$ is the observation noise vector.

It is supposed that the SINS's speed/position information outputs are as follows:

$$
\begin{cases}
V_{SE} = V_{ET} + \delta V_{SE} \\
V_{SN} = V_{NT} + \delta V_{SN} \\
V_{SU} = V_{UT} + \delta V_{SU}
\end{cases}
\tag{7}
$$

$$\begin{cases} L_S = L_T + \delta L_S \\ \lambda_S = \lambda_T + \delta \lambda_S \\ h_S = h_T + \delta h_S \end{cases} \tag{8}$$

where $V_{SE}$, $V_{SN}$, and $V_{SU}$ are the eastward, northward, and upward speeds of the carrier calculated by the SINS, respectively. $V_{ET}$, $V_{NT}$, and $V_{UT}$ are the actual eastward, northward, and upward speeds of the carrier, respectively. $\delta V_{SE}$, $\delta V_{SN}$, and $\delta V_{SU}$ are the eastward, northward, and upward speed errors of the SINS, respectively. $L_S$, $\lambda_S$, and $h_S$ are the latitude, longitude, and altitude of the carrier calculated by the SINS, respectively. $L_T$, $\lambda_T$, and $h_T$ are the actual latitude, longitude, and altitude of the carrier, respectively. $\delta L_S$, $\delta \lambda_S$, and $\delta h_S$ are the latitude, longitude, and altitude errors of the SINS, respectively.

It is supposed that the speed/position information output by the GPS is as follows:

$$\begin{cases} V_{GE} = V_{ET} + \delta V_{GE} \\ V_{GN} = V_{NT} + \delta V_{GN} \\ V_{GU} = V_{UT} + \delta V_{GU} \end{cases} \tag{9}$$

$$\begin{cases} L_G = L_T + \delta L_G \\ \lambda_G = \lambda_T + \delta \lambda_G \\ h_G = h_T + \delta h_G \end{cases} \tag{10}$$

where $V_{GE}$, $V_{GN}$, and $V_{GU}$ are the eastward, northward, and upward speeds of the carrier calculated by the GPS, respectively. $\delta V_{GE}$, $\delta V_{GN}$, and $\delta V_{GU}$ are the eastward, northward, and upward speed errors of the GPS, respectively. $L_G$, $\lambda_G$, and $h_G$ are the latitude, longitude, and altitude of the carrier calculated by the GPS, respectively. $\delta L_G$, $\delta \lambda_G$, and $\delta h_G$ are the latitude, longitude, and altitude errors of the GPS, respectively.

The system observation vector is as follows:

$$\boldsymbol{Z}(t) = \boldsymbol{H}(t)\boldsymbol{X}(t) + \boldsymbol{V}(t) = \begin{bmatrix} \delta V_{GE} - \delta V_{SE} \\ \delta V_{GN} - \delta V_{SN} \\ \delta V_{GU} - \delta V_{SU} \\ \delta L_G - \delta L_S \\ \delta \lambda_G - \delta \lambda_S \\ \delta h_G - \delta h_S \end{bmatrix} \tag{11}$$

Without considering the control action, it is supposed that the stochastic linear discrete system's equation is as follows [22,23]:

$$\begin{cases} \boldsymbol{X}_k = \boldsymbol{\Phi}_{k,k-1}\boldsymbol{X}_{k-1} + \boldsymbol{\Gamma}_{k,k-1}\boldsymbol{W}_{k-1} \\ \boldsymbol{Z}_k = \boldsymbol{H}_k\boldsymbol{X}_k + \boldsymbol{V}_k \end{cases} \tag{12}$$

where $\boldsymbol{X}_k$ is the system's $n$-dimensional state matrix. $\boldsymbol{\Phi}_{k,k-1}$ is the system's $n \times n$-dimensional state transition matrix and can be obtained by the discretization of $\boldsymbol{F}(t)$. $\boldsymbol{\Gamma}_{k,k-1}$ is the $n \times p$-dimensional noise input matrix, and can be obtained by the discretization of $\boldsymbol{G}(t)$. $\boldsymbol{Z}_k$ is the system's $m$-dimensional observation sequence, $\boldsymbol{H}_k$ is the $m \times n$-dimensional observation matrix, $\boldsymbol{V}_k$ is $m$-dimensional observation noise sequence, and $\boldsymbol{W}_{k-1}$ is the system's $p$-dimensional process noise sequence. Simultaneously, $\boldsymbol{W}_k$ and $\boldsymbol{V}_k$ satisfy the following conditions: $E[\boldsymbol{W}_k] = 0$, $Cov[\boldsymbol{W}_k, \boldsymbol{W}_j] = E[\boldsymbol{W}_k\boldsymbol{W}_j^T] = \boldsymbol{Q}_k\delta_{kj}$, $E[\boldsymbol{V}_k] = 0$, $Cov[\boldsymbol{V}_k, \boldsymbol{V}_j] = E[\boldsymbol{V}_k\boldsymbol{V}_j^T] = \boldsymbol{R}_k\delta_{kj}$, and $Cov[\boldsymbol{W}_k, \boldsymbol{V}_j] = E[\boldsymbol{W}_k\boldsymbol{V}_j^T] = 0$. $\boldsymbol{Q}_k$ is the system's noise variance matrix, and $\boldsymbol{R}_k$ is the measurement noise variance matrix.

The KF prediction and update processes are then as follows:

(1) State one-step prediction

$$\hat{\boldsymbol{X}}_{k,k-1} = \boldsymbol{\Phi}_{k,k-1}\hat{\boldsymbol{X}}_{k-1} \tag{13}$$

(2) State estimation

$$\hat{\boldsymbol{X}}_k = \hat{\boldsymbol{X}}_{k,k-1} + \boldsymbol{K}_k(\boldsymbol{Z}_k - \boldsymbol{H}_k\hat{\boldsymbol{X}}_{k,k-1}) \tag{14}$$

(3)　Filtering gain

$$K_k = P_{k,k-1}H_k{}^T(H_kP_{k,k-1}H_k{}^T + R_k)^{-1} \tag{15}$$

(4)　One-step prediction mean square error

$$P_{k,k-1} = \Phi_{k,k-1}P_{k-1}\Phi_{k,k-1}^T + \Gamma_{k,k-1}Q_{k-1}\Gamma_{k,k-1}^T \tag{16}$$

(5)　Estimated mean square error

$$P_k = (I - K_kH_k)P_{k,k-1} \tag{17}$$

where $\hat{X}_{k-1}$ is the estimated state matrix at $t_{k-1}$. $P_{k-1}$ is the error covariance matrix of the optimal filter value at $t_{k-1}$. $Q_{k-1}$ is the system noise variance matrix at $t_{k-1}$. $I$ is a unit matrix.

## 3. ML-aided Methodology during GPS Outages

### 3.1. Proposed System Structures

A novel ML-aided methodology is proposed and was introduced into the SINS/GPS integrated navigation systems during GPS outages, shown in Figure 1.

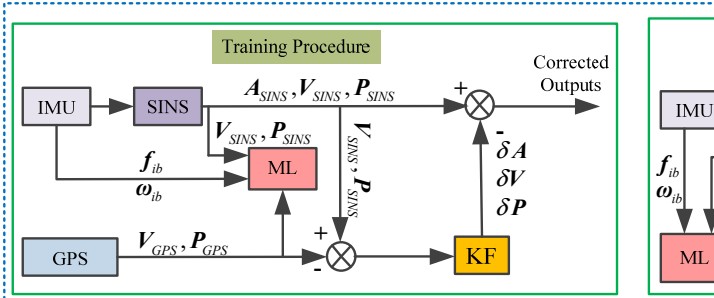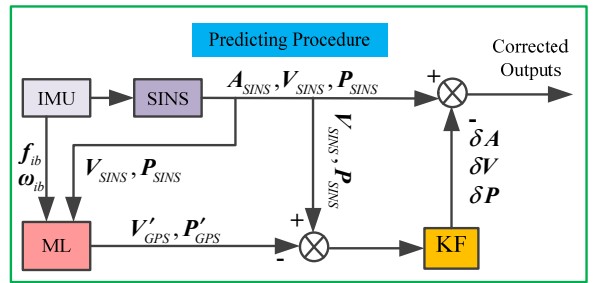

**Figure 1.** Schematic diagram of ML-aided SINS/GPS integrated navigation systems. ML is a machine learning module; KF is a Kalman filtering module; $A_{SINS}$, $V_{SINS}$, and $P_{SINS}$ are the SINS' attitude, speed, and position, respectively; $V_{GPS}$ and $P_{GPS}$ are the GPS' speed and position, respectively; and $V'_{GPS}$ and $P'_{GPS}$ are the speed and position of pseudo GPS information forecasted using the ML method, respectively.

The proposed ML-aided methodology operates as follows. The differences between the speeds/positions of the SINS and GPS are input into the KF module as a measurement value; the estimated attitude, speed, and position errors are fed back to correct the SINS. The SINS's accelerometer, gyroscope, and the speed/position information of SINS are input into the ML module. After a certain amount of data are stored, the data set is trained using a specific machine learning algorithm to obtain the mapping model between the SINS' acceleration, specific force, and speed/position increments outputs, and GPS' speed/position increment. When the GPS signal is unavailable, SINS' navigation data are input into a trained ML model. The model's output is used as the optimal estimation of the actual navigation output. In the training procedure, the accelerometer, gyroscope, and speed/position information of the SINS and GPS were input into the ML module as predictors. The obtained mapping model was as the target variable. In the prediction procedure, the SINS' accelerometer, gyroscope, and speed/position information of the SINS were input into as the ML module as predictors. The estimated pseudo speed/position of the GPS was as the target variable.

### 3.2. The Typical ML Algorithm

DOS-ELM [24] is a newly proposed single hidden-layer feedforward neural network. The model's updating error functions as the forgetting factor's adjustment signal using the DOS-ELM algorithm. If the accuracy rate drops after the model is updated, the algorithm uses a custom formula to reduce the value of the forgetting factor according to the magnitude of the decline. Compared with the backpropagation neural network (BPNN), DOS-ELM has a higher learning speed, and its nonlinear approximation ability is not reduced by this optimization method. The DOS-ELM algorithm's forgetting factor can be automatically and dynamically adjusted according to the iterative error, thus avoiding instability [24]. Therefore, the DOS-ELM algorithm was chosen to aid SINS/GPS integrated systems during GPS outages.

The main steps of the DOS-ELM algorithm can be summarized as follows: a training dataset $D = \{(x_i, t_i)\}_{i=1}^N \subset R^n \times R$, the activation function is $G(\omega, x, b)$, the number of hidden layer neurons is $L$, and the initial forgetting factor $\lambda = 1$.

Step 1. Initialization Phase:

An initial model is obtained by training with the initial training set $D_0 = \{(x_i, t_i)\}_{i=1}^{N_0}$ using the ELM algorithm [25]. Here, the output weight matrix of this initial model is written as $\beta_0$, and the output matrix of the hidden layer is written as $M_0$. The data block identifier is set as $k = 0$, and a transition variable $P_0 = (M_0^T M_0)^{-1}$. The initial model's accuracy using the current initial training set is calculated and labeled $ACC_0$.

Step 2. Online Sequential Learning Phase:

(1)    When a new data block $D_{k+1} = \{(x_i, t_i)\}_{i=(\sum_{j=0}^k N_j)+1}^{\sum_{j=0}^{k+1} N_j}$ identified as $(k+1)_{th}$ begins

processing, the output matrix of the model's hidden layer is updated as follows:

$$M_{k+1} = [\lambda M_k^T \quad M_{k+1}^T]^T \tag{18}$$

where $M_{k+1}$ denotes the output matrix of the hidden layer corresponding to the new data block.

(2)    The model's output weight matrix at the moment is calculated as follows:

$$\beta_{k+1} = \beta_k + P_{k+1} M_{k+1}^T (T_{k+1} - M_{k+1}\beta_k) \tag{19}$$

where $P_{k+1} = \lambda^{-2}P_k - \lambda^{-4}P_k M_{k+1}^T (I + \lambda^{-2}M_{k+1}P_k M_{k+1}^T)^{-1}M_{k+1}P_k$. $T_{k+1}$ denotes the labels of the new dataset, and $T_{k+1} = M_{k+1}\beta_{k+1}$.

(3)    The prediction accuracy of the current model using the new data block is calculated and labelled $ACC_{k+1}$. The accuracy difference between the current model and the model before it was updated is compared as follows:

$$E = ACC_{k+1} - ACC_k \tag{20}$$

(4)    The forgetting factor is updated as follows [25]:

$$\begin{cases} \lambda = \lambda - \frac{1}{5\pi}atan(E) \\ s.t. \ if \ \lambda > 1 \ then \ \lambda = 1 \end{cases} \tag{21}$$

where $atan()$ is an arctangent function. $\lambda \in [0, 1]$, and $\lambda = 1$ indicates that the importance of new data is the same as that of historical data. $\lambda < 1$ indicates that the relative importance of historical data is lower than that of furture data.

The DOS-ELM algorithm takes the model's update error as the adjustment signal for the forgetting factor. If the accuracy rate decreases after the model is updated, the algorithm will use the above equation to reduce the value of the forgetting factor according to the

extent of the decline. This means that the volume of historical information is reduced and the importance of new data is relatively increased, and vice versa.

(5) Check if there are any new data that have not been trained. If so, set $k = k + 1$, return to (1) in Step 2, and the model's training continues. Otherwise, the model training is stopped, and the model parameters are output. At this point, the DOS-ELM-aided methodology for SINS/GPS integrated navigation systems during GPS outages has been implemented.

## 4. Simulation Results

To verify the feasibility and effectiveness of the proposed methodology, actual vehicle-mounted experimental data were used in an offline semi-physical simulation. A SINS prototype, with fiber-optic gyroscopes and quartz accelerometers as its sensors, was used in the experiment. The specific parameters of the inertial measurement unit (IMU) and GPS are shown in Table 1. The vehicle-mounted experiment used the PHINS [26] developed and produced by the French iXBlue company. The prototype's IMU was fixed to a transition board and placed inside the experimental vehicle. The PHINS was set to GPS integration. The attitude, speed, and position information output after integrating the PHINS and GPS were used as the reference for vehicle navigation information. The installation and structure diagrams of the vehicle experiment are shown in Figures 2 and 3, respectively.

**Table 1.** Specific of IMU and GPS parameters.

| Sensors | Parameters | Accuracy |
| :---: | :---: | :---: |
| IMU | Gyroscope Constant Drift | $0.02°/h$ |
| | Gyroscope Random Drift | $0.02°/\sqrt{h}$ |
| | Accelerometer Constant Bias | 50 µg |
| | Accelerometer Random Walk | 50 µg |
| | Frequency | 200 Hz |
| GPS | Position | 1 m |
| | Frequency | 1 Hz |

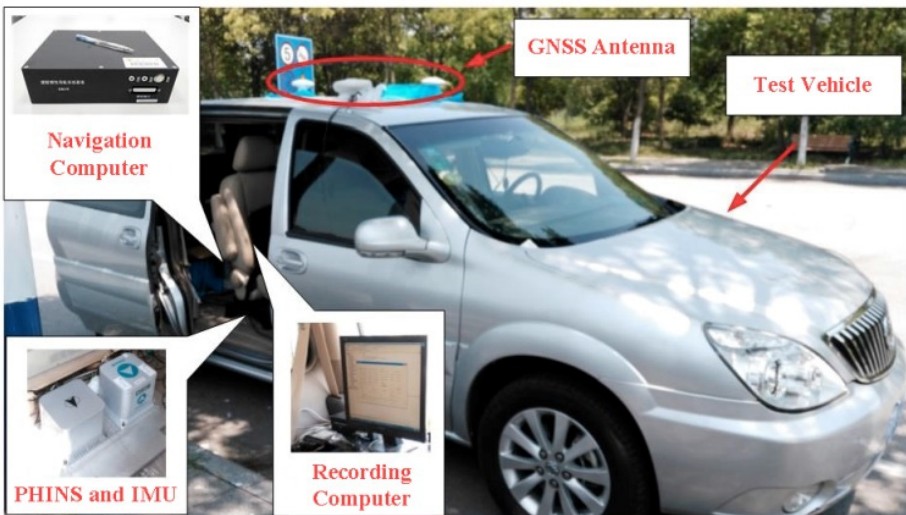

**Figure 2.** Installation diagram for vehicle-mounted experiment.

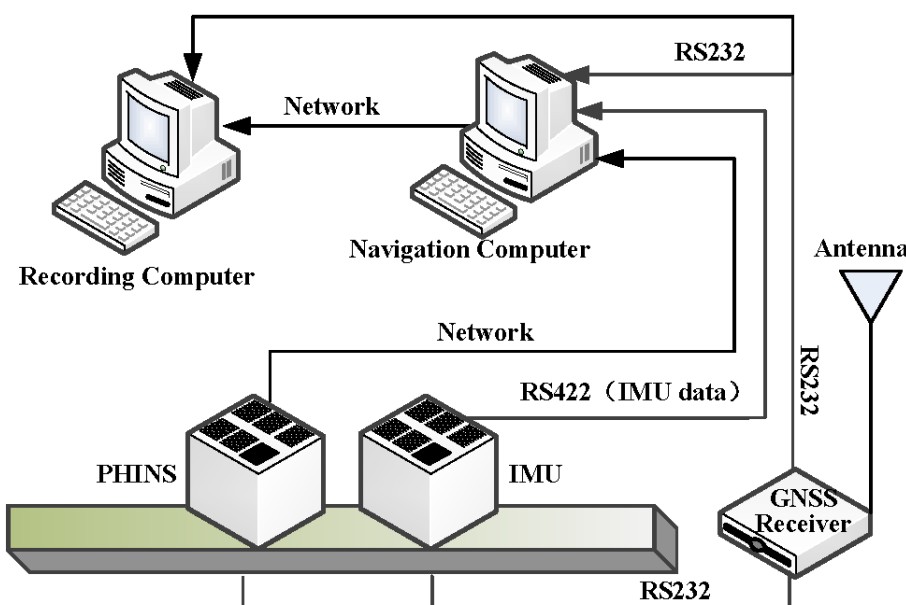

**Figure 3.** Structure diagram of vehicle-mounted experiment.

The whole experiment lasted approximately 5700 s; initial SINS alignment occurred during seconds 0–900, and SINS/GPS loosely integrated navigation occurred during seconds 1800–5700. Data for seconds 1100–3100 were stored as the training data set for the DOS-ELM algorithm. When training was finished, the system entered the prediction phase. Three stages of GPS outages (from the 3500 s to 3800 s; 4100 s to 4300 s; and 4500 s to 4700 s) were artificially set. The navigation track and experimental vehicle trajectory are shown in Figures 4 and 5, respectively. Here, three sections of GPS outages are marked by the red lines.

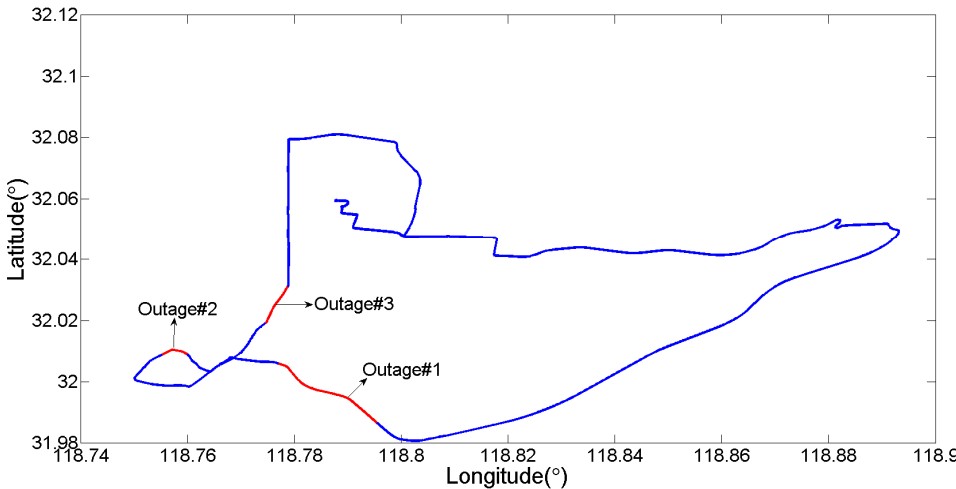

**Figure 4.** Navigation track.

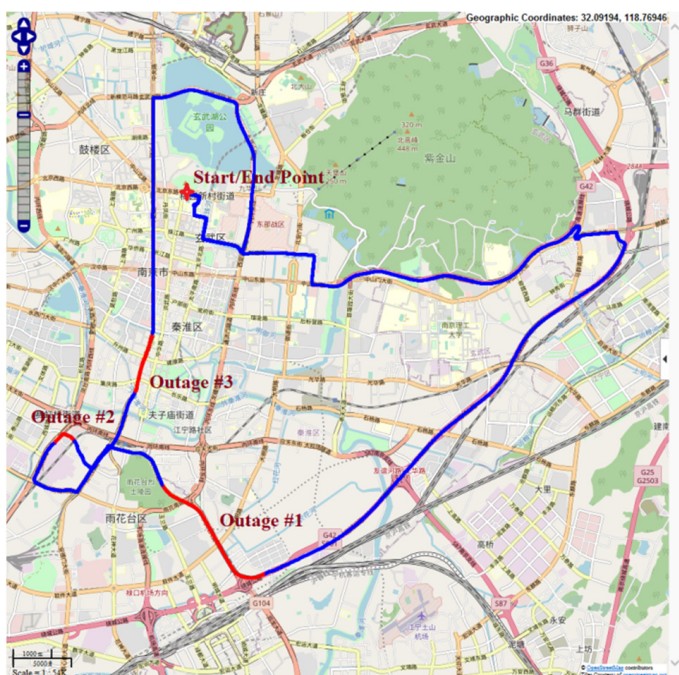

**Figure 5.** The vehicle's experimental trajectory.

The vehicle's dynamic characteristics during the experiment are shown in Figure 6, which illustrates that the vehicle's driving state is complex and repetitive, which met the requirements for ML algorithms training data sets. Figure 7 presents an intelligent estimation process of the pseudo GPS speed/position. The predicted values were very close to the actual speed/position information. To demonstrate the proposed method's advantage, it was compared with the pure INS method. During GPS outages, the system operated in pure INS mode.

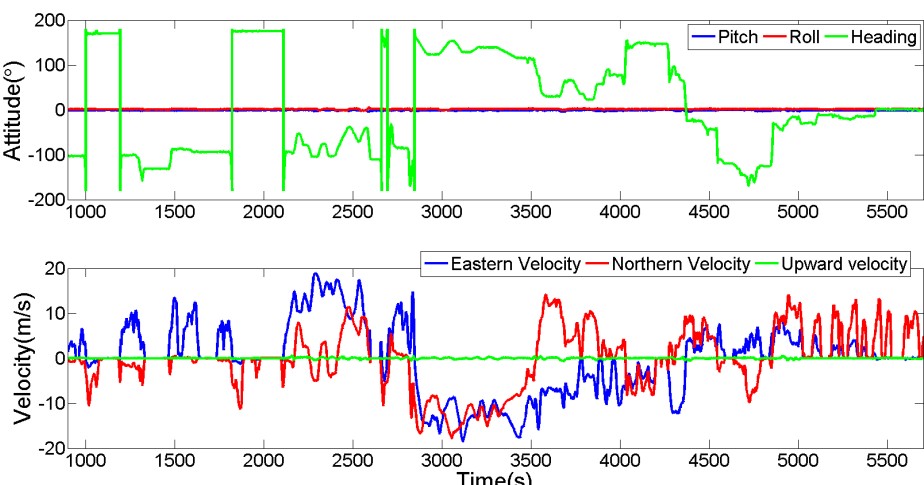

**Figure 6.** The vehicle's dynamic characteristics during the experiment.

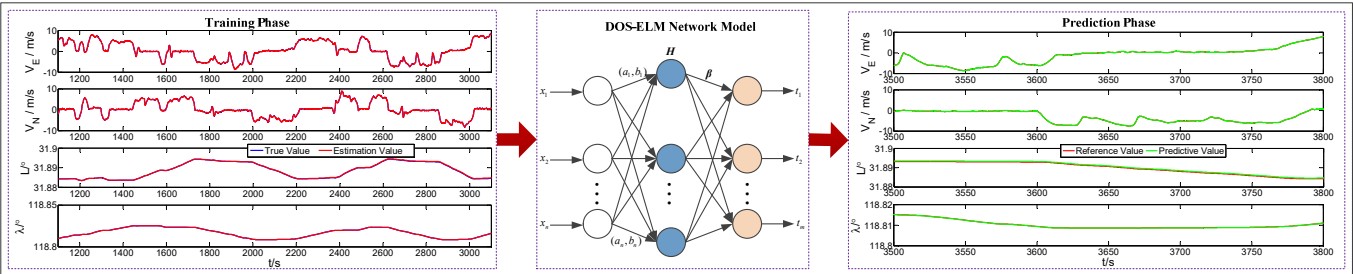

**Figure 7.** Intelligent estimation process of pseudo GPS speed/position.

Figures 8–10 show the speed/position errors of both east and north directions of the DOS-ELM-aided method, the pure INS method, and the high-precision reference during GPS outages #1, #2, and #3, respectively. In these three figures, the red and blue lines correspond to the results of the pure INS and DOS-ELM-aided methods, respectively. The error of the pure INS method generally oscillated sharply and quickly diverged. Although the error correction of the DOS-ELM-aided method to the navigation solution could not achieve an effect similar to the complete convergence of the filter on the error, it dramatically reduced the error value. Table 2 presents the mean and standard deviation (SD) of speed/position errors during GPS outages #1, #2, and #3.

Table 2's comparison of the information fusion algorithm using the DOS-ELM-aided and pure INS methods shows that the DOS-ELM-aided algorithm's navigation calculation error was lower than the pure INS calculation error. The DOS-ELM-aided algorithm's calculation result was very close to the reference value of the integrated system navigation solution. The error values of the pure INS solution were different during GPS outages #1, #2, and #3, indicating that, in the several calculation epochs before the GPS signal failure, the filtering estimation had other correction effects on the INS navigation calculation under normal GPS measurement update conditions. In addition, integrated factors, such as the uncertainty of the IMU device's error drift, affected the GPS. When the GPS signal failed, the INS calculation error was uncertain, whereas the DOS-ELM-aided method realized a partial correction of the INS navigation calculation error.

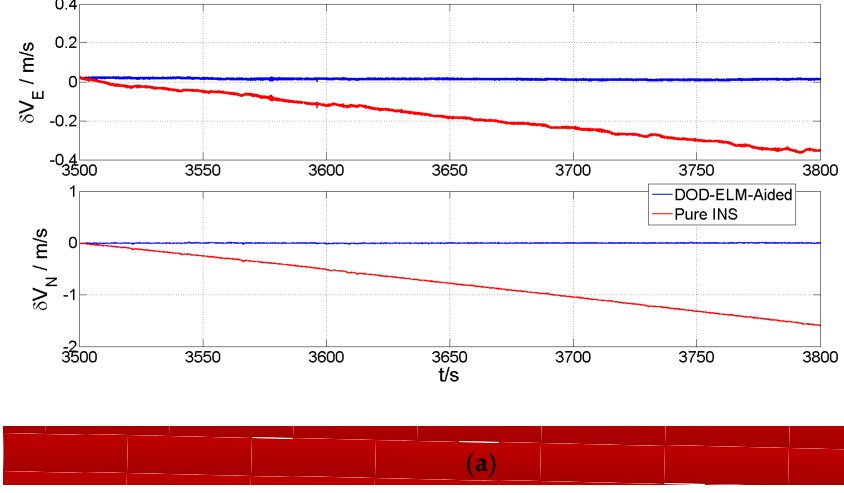

(a)

**Figure 8.** *Cont.*

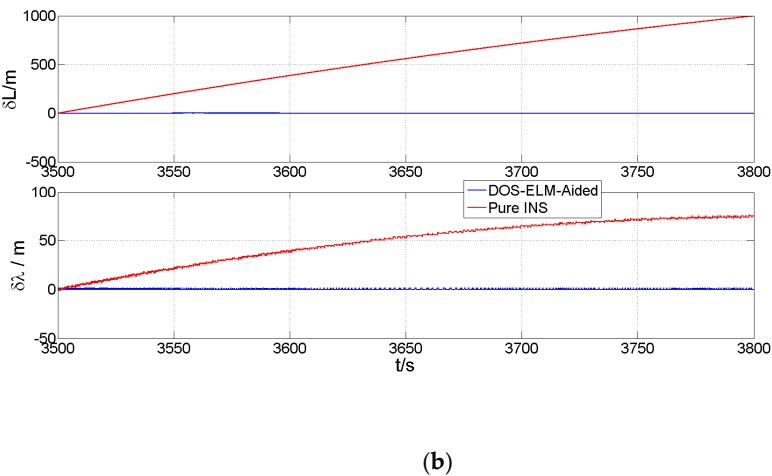

(**b**)

**Figure 8.** Comparison of speed/position errors between DOS-ELM-aided and pure INS methods during GPS outage #1 in the vehicle-mounted experiment. (**a**) Eastern and northern speed errors of DOS-ELM-aided and pure INS methods during GPS outage #1 in the vehicle-mounted experiment. (**b**) Eastern and northern position errors of DOS-ELM-aided and pure INS methods during GPS outage #1 in the vehicle-mounted experiment.

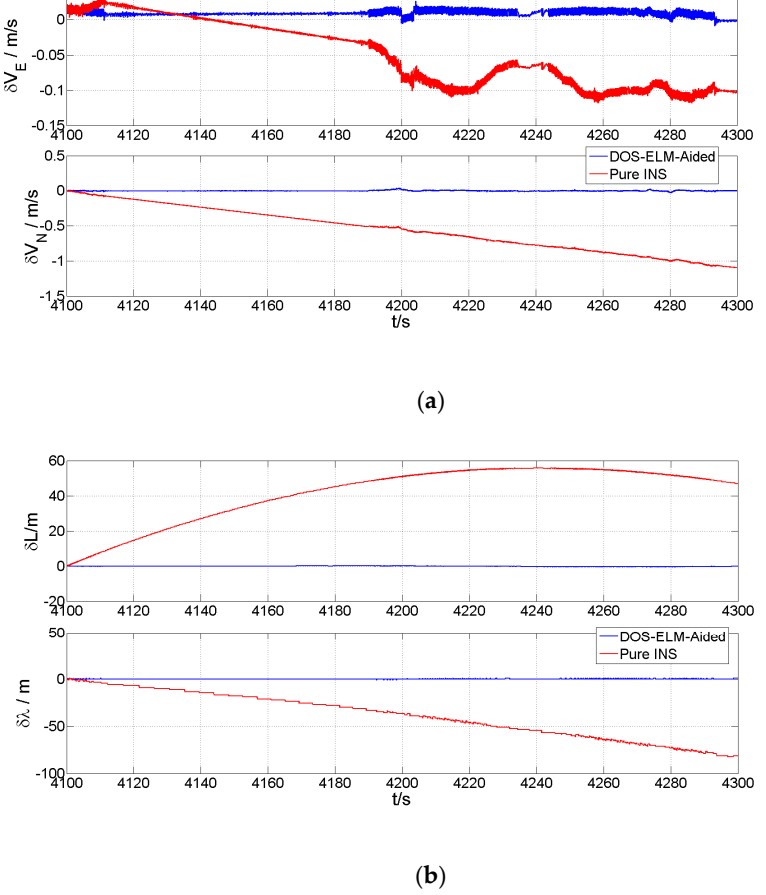

(**a**)

(**b**)

**Figure 9.** Comparison of speed/position errors between DOS-ELM-aided and pure INS methods during GPS outage #2 in the vehicle-mounted experiment. (**a**) Eastern and northern speed errors of DOS-ELM-aided and pure INS methods during GPS outage #2 in the vehicle-mounted experiment. (**b**) Eastern and northern position errors of DOS-ELM-aided and pure INS methods during GPS outage #2 in the vehicle-mounted experiment.

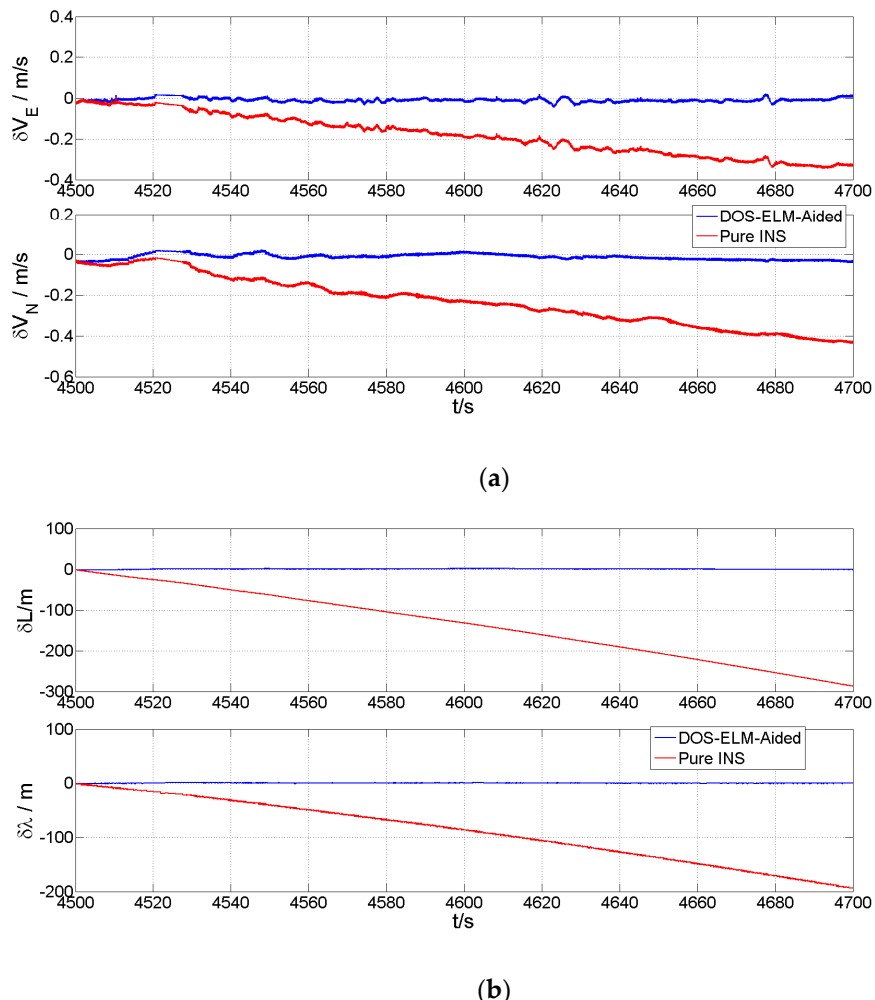

**Figure 10.** Comparison of speed/errors between DOS-ELM-aided and pure INS methods during GPS outage #3 in the vehicle-mounted experiment. (**a**) Eastern and northern speed errors of DOS-ELM-aided and pure INS methods during GPS outage #3 in the vehicle-mounted experiment. (**b**) Eastern and northern position errors of DOS-ELM-aided and pure INS methods during GPS outage #3 in the vehicle-mounted experiment.

**Table 2.** Navigation results of pure INS and DOS-ELM-aided methods.

| Time Periods (s) | Errors | Pure INS | | DOS-ELM/KF | |
|---|---|---|---|---|---|
| | | **Mean** | **SD** | **Mean** | **SD** |
| 3500–3800 | $\delta V_E$ (m/s) | −0.1765 | 0.01082 | 0.01456 | 0.004233 |
| | $\delta V_N$ (m/s) | 0.4611 | 0.4611 | −0.002335 | 0.004276 |
| | $\delta L$ (m) | 289.3 | 289.3 | −0.1713 | 0.1351 |
| | $\delta \lambda$ (m) | 21.99 | 21.99 | 0.1889 | 0.4178 |
| 4100–4300 | $\delta V_E$ (m/s) | 0.3112 | 0.3112 | −0.0008012 | 0.005923 |
| | $\delta V_N$ (m/s) | 0.04615 | 0.04615 | 0.009456 | 0.003912 |
| | $\delta L$ (m) | 15.75 | 15.75 | −0.1452 | 0.1816 |
| | $\delta \lambda$ (m) | 23.97 | 23.97 | −0.08098 | 0.2943 |
| 4500–4700 | $\delta V_E$ (m/s) | 0.1217 | 0.1217 | −0.01169 | 0.014 |
| | $\delta V_N$ (m/s) | 0.09978 | 0.09978 | −0.009198 | 0.009154 |
| | $\delta L$ (m) | 82.41 | 82.41 | 0.3404 | 0.7681 |
| | $\delta \lambda$ (m) | 56.43 | 56.13 | 0.01158 | 0.561 |

## 5. Conclusions

In this paper, a novel DOS-ELM-aided methodology for SINS/GPS integrated navigation systems was proposed to improve navigation accuracy for land vehicles during GPS outages. Data from an actual road vehicle experiment were collected for simulation experiments to verify the feasibility and effectiveness of the proposed methodology. The results showed that the values predicted using DOS-ELM-aided methodology were very close to the actual speed/position information. The proposed method could reduce the divergence of inertial navigation errors and achieve higher positioning accuracy compared to the pure INS algorithm during GPS outages.

In future research, we will investigate the influence of the number of satellites on the DOS-ELM-aided SINS/GPS integrated navigation system. Meanwhile, we will strive to carry out real-time vehicle experiments to make the proposed methodology available in practice.

**Author Contributions:** Conceptualization, J.S.; methodology, J.S.; software, Z.C. and F.W.; validation, J.S., F.W. and Z.C.; formal analysis, Z.C.; investigation, F.W.; writing—original draft preparation, J.S.; writing—review and editing, J.S.; funding acquisition, J.S. All authors have read and agreed to the published version of the manuscript.

**Funding:** This work was partially supported by the National Nature Science Foundation of China under Grant 62203231, the Nature Science Foundation of Jiangsu Province under Grant BK20200763, the State Key Laboratory of Ocean Engineering (Shanghai Jiao Tong University) under Grant GKZD010084, the China Postdoctoral Science Foundation under Grant 2020M681685, the Post-doctoral Research Funding Project of Jiangsu Province under Grant 2021K161B, the Natural Science Research Project of Jiangsu Higher Education Institutions under Grant 19KJB510052.

**Conflicts of Interest:** The authors declare no conflict of interest.

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
