# Peer review of "A Novel ML-Aided Methodology for SINS/GPS Integrated Navigation Systems during GPS Outages"

_remotesensing, doi:10.3390/rs14235932_

Round 1

Reviewer 1 Report

In this paper, a novel ML-aided methodology for the SINS/GPS integrated systems is proposed to improve the navigation accuracy for land vehicles during global positioning system (GPS) outages. The actual sensor data of the land vehicle field experiment is collected online, and then the DOS-ELM-aided methodology for the SINS/GPS integrated navigation systems is applied. It can be seen from simulation results that the proposed methodology can reduce the degree of system error divergence, and then accurate and reliable navigation information can be obtained during GPS outages.

Generally, this topic is very interesting and suitable for publication in Remote Sensing, and the paper is very well written and structured, however, there are still some minor issues that need to be deal with.

1.      In Eq(6) and Eq(12), the meaning of ,  and should be given.

2.      In n frame, namely E-N-U, so in this paper, easting should be eastward, northing should be northward, vertical should be upward.

3.      In this paper, some words should be unified, for example, height or altitude?

4.      Eq (9) may be wrong,   or ?

5.      In , , , , , should be .

6.      Three stages of GPS outages from the 3500 s to 3800 s, 4100 s to 4300 s, and 4500 s to 4700 s are artificially set. Why do you set for this? You should explain in the paper.

7.      The actual vehicle-mounted experimental data is used for the offline semi-physical simulation, whether it can work online, namely carry out the experiment in real time.

Author Response

Dear Reviewer:

 We would  like to express our appreciation to you for providing us with valuable comments for improving the manuscript. We have studied the comments and have made major revisions.

In the following, we present a point-by-point reply to your comments. To make it easier to read, the comments from each reviewer are in blue. The main corrections in the paper are highlighted with yellow. We hope you find our revisions satisfactory. If there are further comments, we are pleased to address them as well.

Your sincerely

Jin Sun, Zhengyu Chen, Fu Wang

Reviewer 2 Report

1. Please the authors add the references for lines 29 to 33, lines 36 to 42, and lines 70 to 81.

2. Lines 109 to 110, “The advantages of this mode are as follows: the integrated mode is simple in structure, and it is easy to implement in engineering.” The advantages of this mode described by the authors are not obvious. The contribution of this mode is not clear. Please clarify this.

3. Please explain the terms and equations in section 2. Please add the process, details, and background for the terms and equations in section 2 to help the readers understand better.

4. In section 3.1, Please explain the methodology of the schematic diagram in figure 1.

5. In section 3.2, what are the predictors and target variable in machine learning? What is the training, validation, and testing set?

6. Lines 329 to 332, “The results showed that DOS-ELM-aided methodology could reduce the divergence of inertial navigation errors and achieve higher positioning accuracy compared to the pure INS algorithm during GPS outages.” Please clarify the novelty and contribution of this manuscript and include more details.

7. What is the limitation and future plan of the manuscript?

8.  The writing of the manuscript must be improved.

Author Response

(The authors gave the same response as above.)

Reviewer 3 Report

See attached file.

Author Response

Dear Reviewer:

We would like to express our appreciation to the reviewers for providing us with valuable comments for improving the manuscript. We have studied the comments and have made major revisions.

In the following, we present a point-by-point reply to your comments. To make it easier to read, the comments from each reviewer are in blue. The main corrections in the paper are highlighted with yellow. We hope you find our revisions satisfactory. If there are further comments, we are pleased to address them as well.

Your sincerely

Jin Sun, Zhengyu Chen, Fu Wang

Round 2

Reviewer 2 Report

Please see the response highlighted in green in the attached file.

Author Response

Dear Reviewer:

We would like to express our appreciation to you for providing us with valuable comments for improving the manuscript again. We have studied the comments and have made major revisions.

In the following, we present a point-by-point reply to your comments. To make it easier to read, the comments from each reviewer are in blue. The main corrections in the paper are highlighted with yellow. We hope you find our revisions satisfactory. If there are further comments, we are pleased to address them as well.

Your sincerely

Jin Sun, Zhengyu Chen, Fu Wang
